

# Whole genome sequencing of *Rhodotorula mucilaginosa* isolated from the chewing stick (*Distemonanthus benthamianus*): insights into *Rhodotorula* phylogeny, mitogenome dynamics and carotenoid biosynthesis

Han Ming Gan[1,2,3], Bolaji N. Thomas[4], Nicole T. Cavanaugh[5], Grace H. Morales[5], Ashley N. Mayers[4], Michael A. Savka[5] and André O. Hudson[5]

[1] Centre for Integrative Ecology-School of Life and Environmental Sciences, Deakin University, Victoria, Australia
[2] Genomics Facility, Monash University, Selangor, Malaysia
[3] School of Science, Monash University, Selangor, Malaysia
[4] College of Health Science and Technology, Rochester Institute of Technology, Rochester, NY, United States of America
[5] Thomas H. Gosnell School of School of Life Sciences, Rochester Institute of Technology, Rochester, NY, USA

Corresponding authors
Han Ming Gan,
han.gan@deakin.edu.au
André O. Hudson, aohsbi@rit.edu

## ABSTRACT

In industry, the yeast *Rhodotorula mucilaginosa* is commonly used for the production of carotenoids. The production of carotenoids is important because they are used as natural colorants in food and some carotenoids are precursors of retinol (vitamin A). However, the identification and molecular characterization of the carotenoid pathway/s in species belonging to the genus *Rhodotorula* is scarce due to the lack of genomic information thus potentially impeding effective metabolic engineering of these yeast strains for improved carotenoid production. In this study, we report the isolation, identification, characterization and the whole nuclear genome and mitogenome sequence of the endophyte *R. mucilaginosa* RIT389 isolated from *Distemonanthus benthamianus,* a plant known for its anti-fungal and antibacterial properties and commonly used as chewing sticks. The assembled genome of *R. mucilaginosa* RIT389 is 19 Mbp in length with an estimated genomic heterozygosity of 9.29%. Whole genome phylogeny supports the species designation of strain RIT389 within the genus in addition to supporting the monophyly of the currently sequenced *Rhodotorula* species. Further, we report for the first time, the recovery of the complete mitochondrial genome of *R. mucilaginosa* using the genome skimming approach. The assembled mitogenome is at least 7,000 bases larger than that of *Rhodotorula taiwanensis* which is largely attributed to the presence of large intronic regions containing open reading frames coding for homing endonuclease from the LAGLIDADG and GIY-YIG families. Furthermore, genomic regions containing the key genes for carotenoid production were identified in *R. mucilaginosa* RIT389, revealing differences in gene synteny that may play a role in the regulation of the biotechnologically important carotenoid synthesis pathways in yeasts.

# INTRODUCTION

*Rhodotorula mucilaginosa* is a common saprophytic fungus that is a part of the Basidiomycota phylum. The organism is typically found in soils, lakes, ocean water, milk and fruit juice (*Wirth & Goldani, 2012*). Of the numerous species in the genus *Rhodotorula,* only *Rhodotorula mucilaginosa*, *Rhodotorula glutinis*, and *Rhodotorula minuta* have been known to be pathogenic to humans (*Wirth & Goldani, 2012*; *Zaas et al., 2003*). Despite being categorized as an opportunistic and emerging pathogen, *R. mucilaginosa* from natural environments appear to possess interesting biological traits ranging from indole acetic acid production (plant growth-promoting), bacterial quorum sensing signal degradation (quorum quenching) to carotenoid production (*Ghani et al., 2014*; *Ignatova et al., 2015*; *Libkind, Brizzio & Broock, 2004*). Despite its genomic potential, resources for *R. mucilaginosa* are surprisingly scarce in public database. To date, the only genomic resource publicly available for this species is from *R. mucilaginosa* strain C2.5t1 that was isolated from the seeds of the cacao plant in Cameroon (*Deligios et al., 2015*). Beyond the NCBI database, another genome of *R. mulaginosa* (strain ATCC58901) can be found in the JGI portal (https://genome.jgi.doe.gov/Rhomuc1/Rhomuc1.home.html) but a user account is required to access the genome.

Carotenoid production in fungi has been suggested as a natural mechanism to protect against photo-oxidative damage in light-intensive environments, given the known antioxidant property of these lipid-soluble pigments as attributed to their chemical structure (*Avalos & Carmen Limon, 2015*; *Cerdá-Olmedo, 1989*; *Echavarri-Erasun & Johnson, 2002*). The biosynthetic pathway of beta-carotene from phytoene has been elucidated in fungal species based on cDNA cloning and enzymatic characterization and was shown to require two major proteins namely, a dehydrogenase and a bifunctional enzyme, encoding both cyclase and phytoene synthase activities (*Sanz et al., 2011*; *Verdoes et al., 2003*). Leveraging on the ease of mutant screening based on visual inspection, the carotenoid pathway in the genus *Rhodotorula* has been conveniently selected for the development of genetic manipulation tool in *Rhodotorula* (*Abbott et al., 2013*; *Koh et al., 2014*; *Sun et al., 2017*) despite the biotechnological significance of this pathway in *Rhodotorula* (*Cutzu et al., 2013*; *Davoli, Mierau & Weber, 2004*; *Libkind, Brizzio & Broock, 2004*; *Marova et al., 2012*; *Taccari et al., 2012*). The heterologous expression of a 3-hydroxy-3-methylglutaryl coenzyme A reductase from *Saccharomyces cerevisiae* substantially increased carotenoid production in *R. mucilaginosa* strain KC8 (*Wang et al., 2017*), indicating the potential of metabolic engineering as alternative and/or complementary approach to growth condition optimization (*Cutzu et al., 2013*; *Davoli, Mierau & Weber, 2004*; *Marova et al., 2012*) for improving carotenoid production in *Rhodotorula* species.

The plant *Distemonanthus benthamianus* is a semi-deciduous perennial tree commonly found in second-growth forests in Nigeria, Cameroon and Ghana (*Adeniyi, Obasi & Lawal,*

*2011*). *D. benthamianus* is of interest given that the plant is used as chewing sticks for dental and oral hygiene by members of Yoruba community in Nigeria. A relatively recent study showed that extracts from the bark of the stems exhibit bactericidal activity against *Staphylococcus aureus* and *Streptococcus mutans*, two bacteria that are often associated with skin and dental infections, respectively (*Adeniyi & Odumosu, 2012*).

In this study, an initial screen for endophytic bacteria that are resistant to the extracts of *D. benthamianus* led to the isolation of a pink-pigmented strain subsequently identified as a fungal strain belonging to the species *Rhodotorula mucilaginosa*. Given the intriguing property of this fungal species and its lack of genomic resources, we sequenced its whole genome on the Illumina platform and performed comparative genomic analysis to gain insight into the carotenoid biosynthetic pathway of this species and more generally the genus *Rhodotorula*. Notably, we also recovered the complete mitochondrial genome of *R. mucilaginosa,* the first for its species and the second for its genus, using genome skimming approach.

## MATERIALS AND METHODS

### Strain isolation

Two grams of internal tissue obtained from surfaced sterilized stem of *Distemonanthus benthamianus* plant was used to inoculate 25 mL of half strength tryptic soy broth (TSB) medium and grown overnight at 30 °C. Microorganisms were isolated by plating 100 μL of 10 fold serial dilutions from $10^{-5}$–$10^{-10}$ of the overnight culture on half strength tryptic soy agar.

### Scanning electron microscopy (SEM)

To fix the organism, 100 μL of cells from $10^{-7}$ dilution from an overnight grown culture was suspended in 3% glutaraldehyde in 0.1 M phosphate buffer pH 7.2 for 30 min. Following fixing, the cells were washed three times and pelleted in sterile water followed by a secondary fixation in 2% osmium tetroxide (in $H_2O$) for 30 min. The cells were washed three more times in sterile water followed by dehydration of the cells in 25%, 50%, 75%, 95% and 100% ethanol for 5 min in each ethanol concentration. The cells were filtered through a 0.22 μm polyethersulfone membrane and incubated at room temperature for 1 h followed by SEM stub mounting and sputter-coating using 10 nm gold/palladium.

### Whole genome sequencing

Total DNA was extracted from a 3-day-old half strength tryptic soy agar culture of *R. mucilaginosa* RIT389 using the MolBio DNA extraction kit according to the manufacturer's instructions. The gDNA was sheared to 500 bp fragment using the Covaris ultrasonicator and subsequently prepared for whole genome sequencing using NEBNext Ultra$^{TM}$ DNA Library Prep kit for Illumina (New England BioLabs, Ipswich, MA, USA). The generated library was subsequently quantified using Qubit and sequenced on the MiSeq (Illumina, San Diego, CA, USA) located at the Monash University Malaysia Genomics Facility using the run configuration of 2 × 250 bp.

## Genome assembly and annotation

Genome size, heterozygosity rate and repeat content were initially estimated using GenomeScope (*Vurture et al., 2017*). Based on the observed high genome heterozygosity of strain RIT389, dipSPAdes version 3.10.1 was used to assemble the whole genome with the additional option of "-expect-rearrangements" activated (*Bankevich et al., 2012*). Genome completeness was calculated using BUSCO3 based on the Basidiomycota odb9 ortholog dataset (*Simao et al., 2015*). Then, gene prediction was performed using GeneMark-ES fungal version (*Borodovsky & Lomsadze, 2011*) with the enhanced intron submodel that can better accommodate sequences with and without branch point sites in the fungal genomes.

Complete mitogenome was recovered by randomly sub-sampling 1/10 of the pair-end reads and assembling them using SPAdes version 3.10.1 (*Bankevich et al., 2012*). The contig corresponding to the whole mitogenome was re-circularized manually, as previously described (*Gan, Schultz & Austin, 2014*) and annotated automatically using MFannot (http://megasun.bch.umontreal.ca/cgi-bin/mfannot/mfannotInterface.pl). Additional genes coding for homing endonucleases commonly found in fungal mitogenomes were identified based on the presence of protein domains corresponding to the GIY-YIG catalytic domain (PF01541.23) and LAGLIDADG endonuclease (PF00961.18, PF03161.12 and PF14528.5) using hmmsearch3 with an $E$-value cutoff of 1e−5 (*Eddy, 2011*).

## Phylogenomics and comparative genomics

Pair-wise average nucleotide identity (ANIm) was calculated using JSpecies (*Richter et al., 2016*) and subsequently visualized with the library package pheatmap in Rstudio. Single-copy genes present in all selected fungal genomes were identified using BUSCO3 (*Simao et al., 2015*). The protein sequences for each ortholog were aligned and trimmed using Muscle and trimAl (-automated1), respectively (*Capella-Gutierrez, Silla-Martinez & Gabaldon, 2009*; *Edgar, 2004*). The final trimmed alignments were concatenated and used to construct a maximum likelihood tree using FastTreeMP (*Price, Dehal & Arkin, 2010*). The reconstructed tree was visualized and annotated using TreeGraph2 (*Stöver & Müller, 2010*).

Identification of proteins involved in the carotenoid biosynthesis pathway was done by scanning the whole predicted proteome for protein domain hits (NC cutoff for TIGRfam and 1e−5 cutoff for Pfam) to lycopene cyclase (TIGR03462, CrtY), phytoene desaturase/dehydrogenase (TIGR02734, CrtI), squalene/phytoene synthase (PF0494, CrtB) and isopentenyl-diphosphate delta-isomerase (TIGR02150). Visualization and comparison of gene neighborhoods were performed using EasyFig with the default BlastN setting (*Sullivan, Petty & Beatson, 2011*). Proteins coded in each genomic sub-region were functionally annotated using Interproscan5 (*Jones et al., 2014*).

## RESULTS AND DISCUSSION

We noticed an organism that was pink/red in color from the screen on tryptic soy agar (Fig. 1A). Based on our previous studies of isolating endophytic organisms, we initially thought the organism belonged to genus *Serratia* or a related genus based on the color of

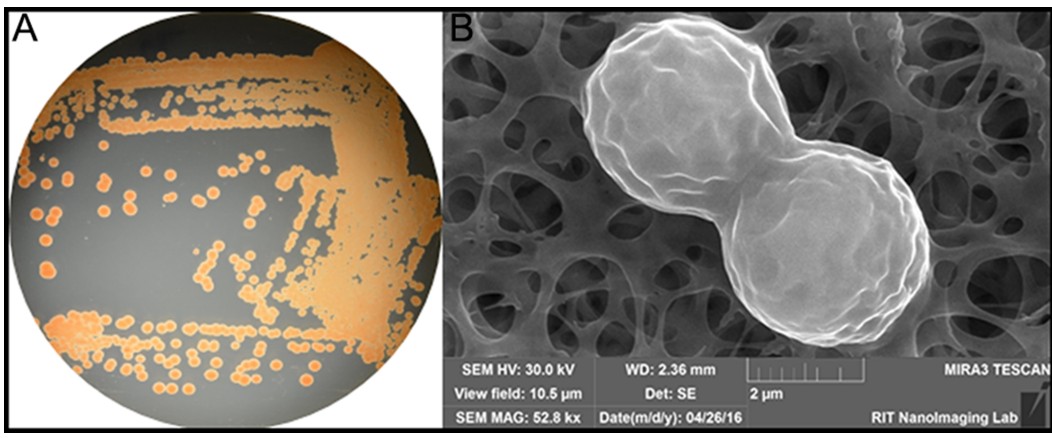

**Figure 1** (A) Color/morphology of *Rhodotorula mucilaginosa* RIT389 grown on half-strength tryptic soy agar (B) Scanning electron microscopy of *Rhodotorula mucilaginosa* RIT389 at 52.8 K magnification.

the colonies. However, based on SEM analysis, it was initially determined that the organism was eukaryotic and not a bacterium based on the size and the morphology depicting cell division (Fig. 1B). The identification of the organism was subsequently confirmed using whole genome nucleotide sequencing.

GenomeScope estimated a genome size of 18.6 mega base pairs (Mbp) with an estimated heterozygosity of 9.29% for strain RIT389 (Fig. 2). The predicted genome size is fairly close to the *de novo* assembled genome length of 19.6 Mbp contained in 250 contigs. The assembled genome has a GC content of 60.28% with an estimated completeness of 89.70%. *De novo* assembly using sub-sampled reads enabled the recovery of the complete mitogenome of strain RIT389 which is the first mitogenome reported for this genus. Approximately 4.55% of the total pair-end reads mapped to the complete mitogenome with an estimated coverage of 400× (Table 1). The complete mitogenome length is 47,023 bp with a GC content of 40.43% which is substantially lower than that of the nuclear genome.

## Genomic and genetic approaches support the species identification of strain RIT389 as *Rhodotorula mucilaginosa*

The ITS region of strain RIT389 exhibits a 100% identity with the sequences of various *Rhodotorula mucilaginosa* strains including the type strain *R. mucilaginosa* ATCC 201848 (Table S1). At the whole genome level, it exhibits the highest average nucleotide identity of 94.10% to strain C2.5t1, the only other genome-sequenced strain of this species at the time of this study (*Deligios et al., 2015*) (Fig. 3). Similar to strain RIT389, strain C2.5t1 is also a plant-associated and was isolated from a cacao seeds (*Theobroma cacao* L) in Cameroon and shown to produce high carotenoid levels when grown in medium supplemented with glycerol (*Cutzu et al., 2013*). Although the plant growth-promoting activity of both strains RIT389 and C2.5t1 has not been studied, a third strain of *R. mucilaginosa*, YR07 isolated from legume plant rhizosphere, has been shown to synthesize up to 45.3 µg of indole acetic

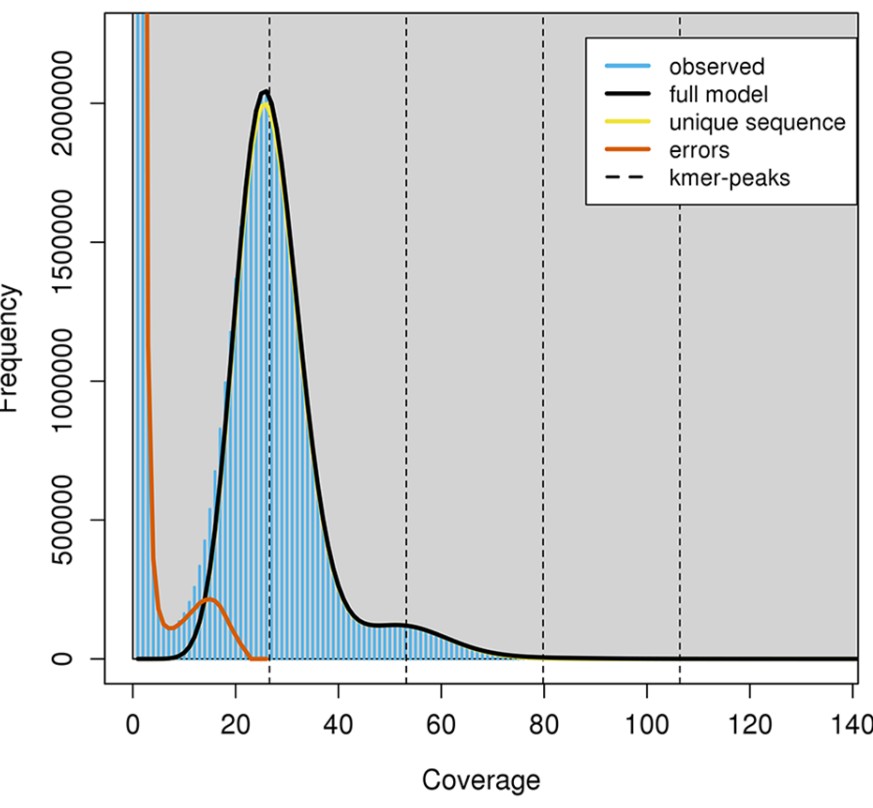

len:18,643,484bp uniq:96.6% het:9.29% kcov:26.6 err:1.06% dup:0.446

**Figure 2** GenomeScope estimation of genome size, repeat content and heterozygosity (Kmer length = 21, Read length = 251 bp and Max kmer coverage = 1,000).

acid (IAA) per mL of culture medium (*Ignatova et al., 2015*). In addition, strain YR07 also exhibits antifungal activity as evidenced by the formation of large inhibition zone against *Fusarium graminearum*, a phytopathogenic fungi.

## Whole genome phylogeny supports the monophyly of the genus *Rhodotorula*

A total of 789 single-copy genes universally present in all 19 fungal strains (Fig. 4) were used to generate a concatenated amino acid alignment consisting of 539,792 sites (234,641 informative sites). By rooting *Microbotyrum violaceum* and *Microbotyrum saponariae* belonging to a different order i.e., Microbotryales, as the outgroup, members of the genus *Rhodotorula* formed a monophyletic group cluster with maximal SH-like support with *R. graminis* WP1 being basal to the rest of the *Rhodotorula* strains (Fig. 4). The lack of strong SH-like support at the shallow relationship especially for members of the species *R. toluroides* is most likely due to the lack of genomic differences, which is expected given that some of the strain names are the alternative strain name of the identical type strain. For example, the type strain designations ATCC10788, IFO0559 and JCM10020 for *R. toluroides* were all derived from the original strain CBS 14. Interestingly, phylogenomic

**Table 1  Strain RIT389 genome statistic and strain information.**

| Organism | *Rhodotorula mucilaginosa* |
|---|---|
| Strain name | RIT389 |
| SRA | SRR5860569 |
| Bioproject | PRJNA390458 |
| Biosample | SAMN07235707 |
| Whole genome: | |
| Accession number | NIUW01000000 |
| Assembled genome length | 19,664,434 bp |
| $N_{50}$ length | 194,287 bp |
| Number of contigs | 250 |
| GC% | 60.28% |
| Predicted protein-coding gene | 7,065 |
| BUSCO Completeness (Basidiomycota odb9) | |
| Complete BUSCOs | 89.70% |
| Complete and single-copy BUSCOs | 86.70% |
| Complete and duplicated BUSCO | 3.00% |
| Fragmented BUSCO | 1.60% |
| Missing BUSCO | 8.70% |
| Total BUSCO groups searched | 1,335 |
| Mitochondrial Genome | |
| Accession number | MF694646 |
| Genome size | 47,023 bp |
| GC% | 40.43% |
| Coverage | 400× |
| Alignment rate | 4.55% |

analysis indicates that the currently sequenced strains of *Rhodotorula toluroides* consist of two major clades with a Jspecies-calculated intraclade and interclade average pair-wise ANI difference of 0.4% and 13%, respectively (Fig. 3). *Rhodotorula* sp. JG1b together with *R. mucilaginosa* strains RIT389 and C2.5t1 formed a monophyletic group that is sister taxa to the major *R. toluroides* group. The close affinity of *Rhodotorula* sp. JG-1b to *R. mucilaginosa* is interesting as it is an eurypsychrophilic yeast isolated from ∼150,000-year-old ice-cemented permafrost soils (*Goordial et al., 2016a*). Given the close genomic affinity of *R.* sp. JG-1b to the currently sequenced *R. mucilaginosa* strains and their diverse isolation source, comparative genomics of these strains may assist in the future identification of novel cold adaptive traits at the molecular level in the genus *Rhodotorula* (*Goordial et al., 2016b*).

## Homing endonuclease-mediated mitogenome expansion in *Rhodotorula mucilaginosa* RIT389

Given the high abundance of mitochondrial organelle in an actively dividing cell, the depth of mitochondrial-derived sequencing reads will be substantially higher than that of the nuclear genome. Thus, by performing a shallow sequencing (genome skimming) on the organism of interest, it is possible to obtain sufficient read representation

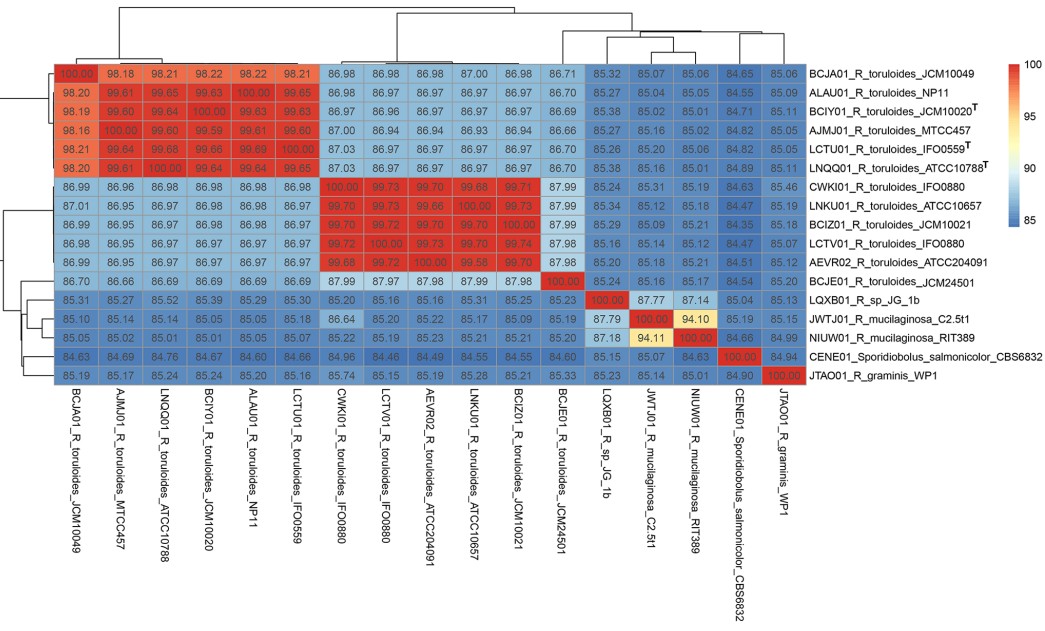

**Figure 3** **Pairwise average nucleotide identity calculation of Rhodotorula genomes.** Genomes with the superscript ''T'' are type strains.

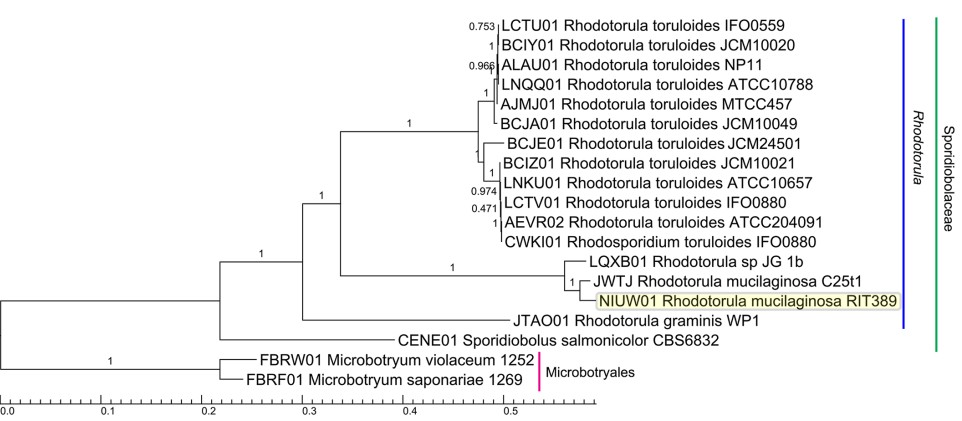

**Figure 4** **Maximum likelihood tree of a concatenated amino acid alignment consisting of 537,792 sites that represent 798 universally present single-copy genes from 19 fungal strains.** Labels on branches indicate shimodaira-hasegawa (SH)-like local branch support values. The scale bar indicates the average number of amino acid substitutions per site.

of the mitochondrial genome which subsequently allows complete assembly. Since the first reported success of genome skimming approach in the construction of the bighorn sheep mitochondrial genome (*Miller et al., 2012*), similar success in recovering mitogenomes across different organisms has been reported (*Froufe et al., 2016*; *Gan et al., 2016*; *Krzeminska et al., 2016*; *Pavlova et al., 2017*). In addition, plastomes and other

**Figure 5 Complete mitochondrial genome of *R. mucilaginosa* RIT389 compared against that of *R. taiwanensis* RS1.** Orange frames indicate coding sequences commonly found in a typical mitochondrial genome. Red and blue arrows indicate transfer and ribosomal RNAs, respectively. Arrow direction represents transcriptional orientation. Dotted lines indicate intronic regions.

high copy number genes have also been routinely recovered and assembled with genome skimming (*Bakker, 2017*; *Gan, Schultz & Austin, 2014*; *Grandjean et al., 2017*; *Richter et al., 2015*; *Straub et al., 2012*).

Contrary to genome skimming, a high coverage whole genome sequencing will generate an extremely high read representation of the mitogenome that can negatively affect mitogenome assembly due to the accumulation of sequencing errors, leading to the generation of fragmented mitogenome (*Mirebrahim, Close & Lonardi, 2015*). In this work, we show that a simple subsampling approach i.e., "*in-silico* genome skimming" followed by *de novo* assembly substantially improves mitogenome assembly and leads to its recovery as a circularized contig in *R. mucilaginosa*.

The reconstructed mitogenome of *R. mucilaginosa* RIT389 is 93% similar to that of *R. taiwanensis* (accession number: HF558455), the only other publicly available complete *Rhodotorula* mitogenome assembled from a low output paired-end run (470 megabases output) of Roche 454 Genome Sequencer (*Zhao et al., 2013*) despite the availability of various *Rhodotorula* whole genome sequences in the public database.

Future study focusing on the reconstruction of complete mitogenome using *in-silico* genome skimming approach from fungal whole genome sequencing data that are publicly available in the the NCBI sequence read archive (SRA) will be instructive.

Despite exhibiting a similar mitochondrial gene arrangement and a relatively high nucleotide sequence similarity to *R. taiwanensis* RS1, the assembled complete mitogenome of strain RIT389 is at least 7,000 bp larger than that of *R. taiwanensis*. Gene neighborhood analysis indicates that a majority of the length difference was largely due to the presence of intronic regions containing homing endonuclease genes (Fig. 5) which is consistent with other studies reporting the prevalence of fungal mitogenome size polymorphism among species from the same genus due to intron acquisition (*Joardar et al., 2012*; *Kanzi et al., 2016*). Homing endonucleases recognize and cleave target sites ranging from 14 to 40 base pairs which match the intron insertion site in donor DNA (*Belfort et al., 2005*). The homing endonucleases identified in both *Rhodotorula* mitogenomes belong to the LADLIDADG and GIY-YIG families. Both LADLIDADG and GIY-YIG endonucleases were named according to the signature motifs presence in their protein sequence. For example, the GIY-YIG endonucleases are characterized by the presence of a structural domain with two short

motifs "GIY" and "YIG" in the N-terminal (*Dunin-Horkawicz, Feder & Bujnicki, 2006*).

In strain RIT389, approximately 3 kilobases of the large mitochondrial ribosomal RNA gene consist of intronic regions coding for LAGLIDADG-type endonuclease and in contrast, these regions are completely absent in the mitogenome of *R. taiwanensis* RS1. A GIY-YIG endonuclease ORF could also be identified within the 1.5 kilobases intronic region of RIT389 *nad5* gene which is absent in that of *R. taiwanensis* RS1. Presence of intronic region in the mitochondrial *nad5* has been previously reported in *Basidiomycota* and Ascomycota species such as *Trametes cingulata, Moniliophthora perniciosa, Ustilago maydis* and *Rhynchosporium commune* (*Abbott et al., 2013*; *Formighieri et al., 2008*; *Haridas & Gantt, 2010*). However, in the reported fungal mitogenomes, the intronic ORF(s) in *nad5* encodes for LAGLIDADG-type endonuclease instead of GIY-YIG endonuclease. The first piece of evidence for a mobile intronic GIY-YIG endonuclease ORF in fungi was demonstrated by the efficient transfer of the GIY-YIG ORF from the second intron of mitochondrial cytochrome b gene in *Podospora curvicolla* to a GIY-YIG-less allele (*Saguez, Lecellier & Koll, 2000*).

## Identification of a genomic region associated with carotenoid biosynthesis

Essential genes required for the biosynthesis of carotenoid could be identified in strain RIT389 which is consistent with its red coloration (Fig. 1A, Table S2), a visual evidence for carotenoid production. Such notable phenotype associated with carotenoid production presents a huge advantage for molecular cloning and characterization of this pathway. As expected, genes involved in the carotenoid synthesis pathway were frequently cloned and characterized from a wide variety of bacteria, archaea, fungi and plants (*Li et al., 2011*; *Li et al., 1996*; *Misawa et al., 1995*; *Nupur et al., 2016*; *Reddy et al., 2017*; *Van Dien et al., 2003*; *Yang et al., 2015*).

The genes coding for phytoene synthase (*crtB*), lycopene cyclase (*crtY*), and phytoene desaturase (*crtI*) are located in relatively close proximity with one another while the gene coding for the enzyme geranyl pyrophosphate synthase which is crucial for the production of an early precursor for carotenoid is located on separate contig (Table S2). As observed in several fungal species, the *crtB* and *crtY* genes are fused and thus code for a bifunctional protein containing both lycopene cyclase and phytoene synthase activities (*Arrach et al., 2001*; *Sanz et al., 2011*). Within the genus *Rhodotorula*, the gene coding for carotenoid oxygenase (*crtX*) responsible for the cleavage of carotenoid to retinal (Vitamin A) and *crtBY* are located in close proximity and are convergently transcribed except in the species *R. mucilaginosa*, whereby *crtX* and *crtBY* are divergently transcribed and are separated by a large gene coding for OPT family small oligopeptide transporter (Fig. 6). In *Fusarium fujikuroi*, mutation in the *crtX* gene led to the overproduction of carotenoid (*Prado-Cabrero et al., 2007*), suggesting that carotenoid oxygenase is involved in the regulation of the carotenoid synthesis through a negative feedback mechanism. The notable difference in gene arrangement and transcription orientation involving *crtX* can therefore affect the regulation of carotenoid synthesis and accumulation in *R. mucilaginosa* (*Noble & Andrianopoulos, 2013*). It is also worth noting that the *crtI* gene in strain RIT389

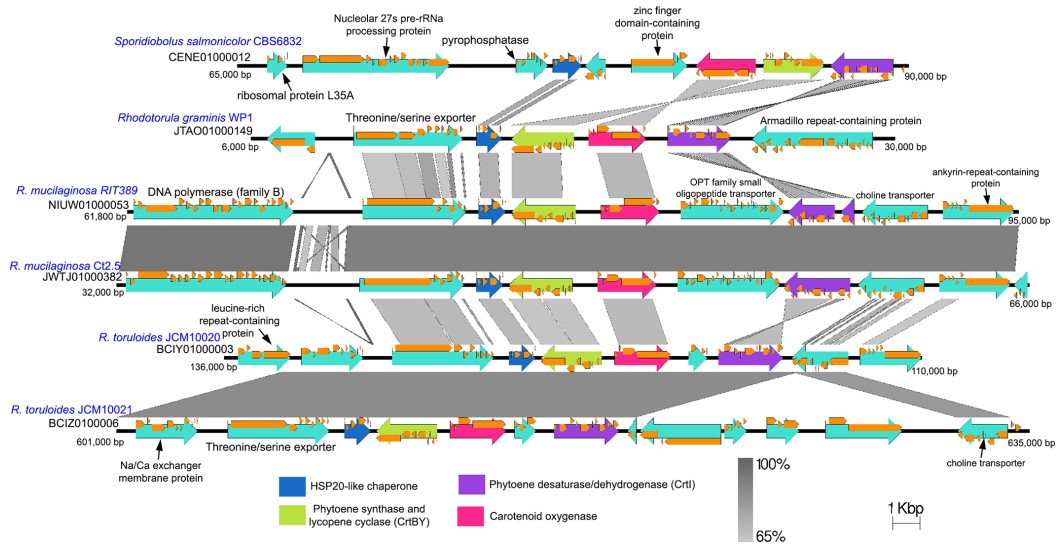

**Figure 6** **Comparison of genomic sub-region containing the gene cluster associated with carotenoid biosynthetic pathway.** Orange frames within the teal arrows indicate the coding sequences in the exonic regions of the corresponding genes.

was predicted as two separate genes which is unexpected given that the gene region exhibits high nucleotide homology and coverage to its respective orthologs in strains WP1 and Ct2.5. Whole transcriptome analysis of strain RIT389 will be necessary to validate the predicted spliced *crtI* gene in the future.

## CONCLUSION

We demonstrate the feasibility of reconstructing the whole genome and complete mitogenome of *Rhodotorula mucilaginosa* using only Illumina short reads. The whole genome of *R. mucilaginosa* is the second to be reported to date for its species. Despite the availability of various whole genome sequences of *Rhodotorula* in public databases, the complete and annotated mitogenome of *Rhodotorula mucilaginosa* strain RIT389 is the first to be successfully reconstructed via *in-silico* genome skimming and annotated for its species. We also highlight the considerable dissimilarity in the syntheny of carotenoid synthesis gene cluster among *Rhodotorula* strains with potential implications in the regulation of carotenoid production.

## ACKNOWLEDGEMENTS

The authors thank Dr. Richard Hailstone from the Chester F. Carlson Center for Imaging Science (RIT) for assistance with SEM analysis.

### Funding

Funding for this work was provided by the following: the College of Science (COS) and the Gosnell School of Life Sciences (GSoLS) at Rochester Institute of Technology (RIT), a Research Laboratory and Faculty Development Award from the College of Health Sciences and Technology (RIT) and the Monash University Malaysia Tropical Medicine and Biology Multidisciplinary Platform. The funders had no role in study design, data collection and analysis, decision to publish or preparation of the manuscript.

### Grant Disclosures

The following grant information was disclosed by the authors:
College of Science (COS).
The Gosnell School of Life Sciences (GSoLS).
College of Health Sciences and Technology (RIT).
Monash University Malaysia Tropical Medicine and Biology Multidisciplinary Platform.

### Competing Interests

The authors declare there are no competing interests.

### Author Contributions

- Han Ming Gan conceived and designed the experiments, performed the experiments, analyzed the data, wrote the paper, prepared figures and/or tables, reviewed drafts of the paper.
- Bolaji N. Thomas and Michael A. Savka conceived and designed the experiments, contributed reagents/materials/analysis tools, wrote the paper, reviewed drafts of the paper.
- Nicole T. Cavanaugh, Grace H. Morales and Ashley N. Mayers performed the experiments, reviewed drafts of the paper.
- André O. Hudson conceived and designed the experiments, analyzed the data, contributed reagents/materials/analysis tools, wrote the paper, prepared figures and/or tables, reviewed drafts of the paper.

### DNA Deposition

The following information was supplied regarding the deposition of DNA sequences:
This Whole Genome Shotgun project has been deposited at DDBJ/ENA/GenBank under the accession NIUW00000000. The version described in this paper is version NIUW01000000.

### Data Availability

GenBank: NIUW01000000.

## Supplemental Information

Supplemental information for this article can be found online at http://dx.doi.org/10.7717/peerj.4030#supplemental-information.

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
