# Peer review of "Whole genome sequencing of Rhodotorula mucilaginosa isolated from the chewing stick (Distemonanthus benthamianus): insights into Rhodotorula phylogeny, mitogenome dynamics and carotenoid biosynthesis"

_PeerJ, doi:10.7717/peerj.4030_

## Round 0.1 · original submission · Minor Revisions

As noted by the two reviewers, the manuscript requires a number of Minor Revisions. Please address all their comments.

·

Basic reporting

The text is well written and the article is consistently written in professional english. A few long sentences may be reframed to simplify the text for readers. Raw data is shared as required and results are well supported with evidences

Experimental design

The methods used are sufficient for the article and the mitochondrial genome elucidation through 'genome skimming' is very innovative. However, I think the readers would like a more detailed account on 'recovery of the complete mitochondrial genome of R. mucilaginosa using the genome skimming approach' as there are few reports on this this and also this is the main attraction of this review. Thee should be more references on similar genome skimming methods used or on other methods for mitochondrial genome recovery. Otherwise, there are several reports on description of the genome and transcriptome of Rhodotorula.

I didnot follow why the authors have focussed on Distemonanthus benthamianus in detail. I think the details should be minimized as the site of isolation has no effect on the description of genome. If there is some significance of this plant in this context then it should be clarified in the article

Validity of the findings

Data is robust and although there is nothing much to be 'newly' reported, the mitochondrial genome recovery does give this article an edge above the other articles/reports published on genome of Rhodotorula.

Additional comments

The results and discussion section and conclusion section can be made more informative by
(1) comparison to other similar reports
(2) more description on genome skimming
(3) comparing the presence of carotenoid pathway gene(s) in yeast Rhodotorula and plants or bacteria, since the article is more focused on carotenoids production and its importance as a valuable anti-oxidant
(4) I believe readers would also want to know details on "homing endonuclease from the LAGLIDADG and GIY-YIG families" and a section can be added on this

Reviewer 2 ·

Basic reporting

The manuscript describes the isolation of a putative endophyte from the plant Distemonanthus benthamianus, its identified as a strain of Rhodotorula mucilaginosa, genome sequencing and analysis of the genome. Primary observations from the genome were the annotation of the mitochondrial genome and identifying the candidate genes for synthesis of carotenoid molecules.

The manuscript is a solid set of data. The following points are aimed to improve the text.

A general point is that the preexisting literature has been someone sparsely covered in places, and that then limits some of the conclusions that can be drawn. For example:

(1) Line 58 species within the genus and “validating the monophyly of the genus Rhodotorula” and line 307. Here consideration of the two Wang et al. 2015 papers in Stud Mycol. 81: 27-53 and 149-89 would be wise, since these were the study to clean up Rhodotorula, as until then the genus was polyphyletic. The conclusion on line 307 is too strong without all the species in the genus being included.
(2) Lines 94-97: a key point is that the genes have been mutated and shown to be involved in synthesis of pigment. This was first done in the closely related genus Sporobolomyces (Abbott et al. 2013 Appl Microbiol Biotechnol (2013) 97: 283-295), and then the two genes in Rhodosporidium (=Rhodotorula) toruloides (Koh et al. 2014 BMC Microbiol 14:50; Sun et al. 2017 Biotechnol Lett 39: 1001-1007).
(3) Strain RIT389 should be deposited in one or more culture collections, e.g. CBS.

Text edits:

Line 83: write as “resources for R. mucilaginosa are surprisingly scare in public databases”.
Lines 83-84: JGI has also sequenced R. mucilaginosa strain ATCC58901. Given the discussion on the genus, it may be worth pointing out that there are genome projects for other Rhodotorula species.
Line 91: “in at least three different fungal species” is too vague.
Lines 107-108: Replace “A relatively recent study showed that” with “The”.
Lines 192-193: how the identification was may should be clearer, i.e. PCR of ITS and sequencing.
Lines 197-198: there appear to be two unrelated points in this sentence, so better split into two.
Line 213: add “levels” for “carotenoid levels when”.
Line 226: “i.e.” in place of “e.g.”.
Lines 233, 235 and 239: spelling of “toruloides”.
Lines 237 and 241: using “Rhodotorula sp.” would be better.
Line 274: “Identification of a genomic region”.
Lines 280 and 281: better with the addition of two commas, as synthase, and carotenoid, .
Line 304: “databases” plural.
References: check for formatting, e.g. italics on species names, remove capital letters in article titles.

Experimental design

Lines 187-192: (a) This section needs more background on the isolation of the microbe from the chewing stick. (b) While the description may have been what happened, it does not present the authors in a strong light. SEM is over-kill to show something is not a bacterium.

Validity of the findings

See comment 1 in section 1 about the claims on monophyly of the genus Rhodotorula.

---

## Round 0.2 · accepted · Accept

Congratulations,

The manuscript can be Accepted for publication.